# Open-Vocabulary Object Detection for Low-Altitude Scenarios Using RGB-Infrared Data: A Benchmark and A New Method

## Abstract

Traditional object detection methods are limited by closed datasets, while open-vocabulary object detection (OVOD) overcomes this limitation. However, most existing OVOD approaches are trained on natural scene images and struggle to generalize to low-altitude scenes images (e.g., UAV-captured images) due to domain differences between the datasets. Therefore, this paper aims to advance research on open-vocabulary object detection in low-altitude scenarios. Unlike most existing open-vocabulary methods, which are trained solely on RGB images and corresponding textual annotations, this paper proposes the first low-altitude open-vocabulary object detection dataset using aligned RGB-Infrared images, named RGB-Infrared OVOD Dataset (RIOVOD), equipped with vocabulary-level annotations. We aim to leverage the complementary information between the two modalities, specifically by utilizing the texture features of RGB images and the thermal radiation information from infrared images, to further enhance the performance of OVOD methods. Building on this, we proposed a new architecture for open-vocabulary object detection in low-altitude scenarios using RGB-Infrared data, which is named LSRI. Extensive experiments show that our method outperforms other approaches, achieving a 0.356 $AP_{50}$ on the RGB-Infrared OVOD Dataset, compared to 0.246 and 0.302 $AP_{50}$ achieved by single-modal (RGB and Infrared) methods, respectively.

## 1 Introduction

Aerial object detection aims to accurately locate and classify targets in aerial images, serving as a foundational technology in key areas like urban planning, environmental monitoring, and disaster response. Although some traditional object detection algorithms, such as Faster R-CNN (Ren et al., 2016) , Yolo (Varghese & Sambath, 2024) and so on, have made significant progress, these methods are heavily reliant on closed-category training data. This limitation results in poor performance when dealing with unknown categories beyond predefined categories, thereby hindering their widespread application in real-world scenarios.

To address this issue (Zhu & Chen, 2024), Open-Vocabulary Object Detection(OVOD) has emerged and gradually become a research hotspot. OVOD addresses this by introducing vision-language models (Radford et al., 2021; Liu et al., 2024b), using natural language as a supervisory signal to align image regions with text embeddings. However, most existing OVOD methods are trained and tested on natural images (Zang et al., 2024), and due to the significant differences between datasets in different domains, these methods struggle to generalize effectively to other tasks in aerial object detection (Pan et al., 2025). Some studies (Li et al., 2024; Zang et al., 2024; Pan et al., 2025) have conducted OVOD based on aerial imagery, achieving good performance. This not only demonstrates the strong generalization potential of OVOD in handling diverse and unknown categories of objects, but also suggests the importance of domain-specific datasets. In contrast, research on OVOD for aerial object detection in low-altitude scenarios is relatively limited, and most of the existing work relies on traditional closed-set object detection methods, which are restricted to detecting targets within a limited set of categories (Wei et al., 2025). Aerial images captured by low-altitude drones share similarities with remote sensing images, but they are collected at lower altitudes and contain more easily recognizable targets (Du et al., 2019). Therefore, applying OVOD techniques for low-altitude drone object detection holds significant research value and application potential (Tian et al.,

2025). Based on this, this paper aims to advance the research on open-vocabulary object detection in low-altitude drone scenarios.

Although OVOD has made some progress in natural scenes, OVOD detection in low-altitude drone scenarios still faces some unique challenges. **Firstly,** targets are relatively small in scale and have complex spatial distributions. In images captured by low-altitude drones, targets often have small spatial scales, and in urban scenes, their distribution is dense and chaotic. Objects such as buildings and vehicles frequently overlap and occlude each other. **Secondly**, target features often degrade, such as low contrast, blurred edges, and texture loss. This significantly reduces the distinguishability between targets and backgrounds or similar categories, limiting the generalization ability of traditional detection methods. **Finally**, most existing methods primarily rely on single-modal data for object detection, utilizing multimodal datasets may be an effective way to mitigate the above issues (Xiong et al., 2025). Multimodal data provides complementary information, which can help address challenges that single-modal data struggles with. However, there is a scarcity of multimodal aerial datasets, which has become a major bottleneck in advancing related research.

To address the above challenges, we have constructed the first alignment-based RGB-Infrared image open-vocabulary object detection dataset and proposed a novel baseline method. Specifically, this paper organizes the DroneVhicle-aligned RGB-Infrared dataset (Sun et al., 2022) to create a dataset (RI-OVOD) suitable for open-vocabulary object detection, ensuring no category overlap between the training and test sets. Furthermore, we present the LSRI model, which is built upon the Decoupled OSOD (DOSOD) architecture (He et al., 2024) and capable of processing multimodal infrared and RGB data for open-vocabulary object detection.

In terms of model design, this paper introduces the following key techniques: First, to enhance the model's discriminative ability for weakly-appearing targets, we propose a general dual-dimension feature enhancement module (DDFE). This module leverages the complementarity of multimodal data by using the feature information from one modality to enhance the features of the other modality in both spatial and channel dimensions. Second, to address the challenge of small target detection in low-altitude UAV scenarios, we introduce $L_{SAFit}$ that adjusts the bounding box loss based on the target size, providing stable and accurate supervision for model training.

Our contributions can be summarized as follows:

- We constructed RI-OVOD Dataset, the first open-vocabulary object detection dataset using aligned RGB-Infrared images, filling the data gap in such tasks under low-altitude UAV scenarios.

- We proposed LSRI, the first model for OVOD using aligned RGB-Infrared images. It introduces DDFE block and the $L_{SAFit}$ loss function, constructing an open-vocabulary detection model with strong generalization ability.

- Experimental results demonstrate that the proposed model outperforms other OVOD methods on RI-OVOD Dataset.

## 2 RELATED WORK

### 2.1 LOW-ALTITUDE DRONE SCENARIO DATASET

Many typical aerial image object detection datasets have been proposed, such as VisDrone (Du et al., 2019), which covers various urban and suburban scenes and annotates target categories such as vehicles, pedestrians, and bicycles; DOTA (Xia et al., 2018), which provides high-resolution rotated bounding box annotations suitable for multi-category general object detection tasks; DIOR (Li et al., 2020) containing 20 common target categories and xView (Lam et al., 2018) containing 60 target categories provide large-scale multi-category support for aerial object detection. Although the above datasets lay the foundation for aerial object detection, they are generally designed for traditional object detection methods and are mostly based on RGB single-modal images. Currently, only a few datasets, such as LAE-1M (Pan et al., 2025) and MI-OAD (Wei et al., 2025) Dataset, support open-vocabulary object detection in the aerial domain. Additionally, publicly available multimodal aerial object detection datasets remain scarce. Existing multimodal datasets, such as DroneVehicle (Sun et al., 2022) and RGB-Tiny (Ying et al., 2025), although covering some basic object categories, still have limitations such as a limited number of target categories and a lack of text annotations

corresponding to target semantics. This limitation significantly hinders the development of open-vocabulary object detection research in the aerial remote sensing field.

## 2.2 OPEN-VOCABULARY OBJECT DETECTION

Open-vocabulary object detection aims to overcome the reliance of traditional object detection methods on predefined categories, leveraging vision-language models to detect unseen category targets. In recent years, Some models like CLIP, RegionCLIP, and DetCLIP (Radford et al., 2021; Zhong et al., 2022; Yao et al., 2022) have significantly improved the recognition of unseen category targets by mapping image region features and textual descriptions into a shared semantic space. GroundingDINO (Liu et al., 2024b) further combined object detection with language grounding tasks, greatly enhancing detection performance in natural image scenes. YOLO-World (Cheng et al., 2024) proposed a re-parameterizable vision-language path aggregation network and designed a region-text contrastive loss, balancing detection accuracy and inference speed.

However, existing OVOD methods are predominantly based on natural image scenes, making it challenging to directly transfer to low-altitude scene. Some studies have attempted to apply OVOD methods to aerial object detection tasks, often relying on pre-trained CLIP models or pseudo-labeling techniques for transfer learning. For example, CastDet (Li et al., 2024) was the first to introduce OVOD to the aerial remote sensing domain, proposing a CLIP-activated teacher-student detection framework and validating its effectiveness on multiple remote sensing datasets. LAE-DINO (Pan et al., 2025) proposed the Locate Anything on Earth (LAE) task, built the first 80-class open-vocabulary object detection dataset for aerial scenes, and provided foundational resources for research in this field.

## 3 RGB-INFRARED OVOD DATASET

Built upon an existing dataset (Sun et al., 2022), we present the first dataset designed for open-vocabulary aerial object detection using aligned RGB-Infrared multimodal imagery: the RGB-Infrared OVOD Dataset. The dataset contains 25,626 aligned RGB-Infrared image pairs and annotations for approximately 340,000 objects, with annotations using vocabulary-level text descriptions. As shown in Table 1, Compared with other existing dataset, our dataset effectively fills the gap in existing low-altitude scene OVOD datasets and provides the foundational dataset for introducing natural language into RGB-Infrared multimodal object detection research.

| Dataset | Datatype | F-OVOD | Images | Instances | Categories |
|---|---|---|---|---|---|
| DIOR Li et al. (2020) | RGB | × | 23 463 | 192 518 | 20 |
| DOTA v2.0 Xia et al. (2018) | RGB | × | 19 871 | 495 754 | 10 |
| Visdrone Du et al. (2019) | RGB | × | 29 040 | 740 419 | 10 |
| DroneVehicle Sun et al. (2022) | RGB-Infrared | × | 56 878 | 939 287 | 5 |
| xView Lam et al. (2018) | RGB | × | 9961 | 732 960 | 60 |
| RGB-Tiny Ying et al. (2025) | RGB-Infrared | × | 93 000 | 1 200 000 | 7 |
| LAE Pan et al. (2025) | RGB | ✓ | - | - | 80 |
| MI-OAD Wei et al. (2025) | RGB | ✓ | 163 023 | 2 000 000 | 100 |
| **RI-OVOD (ours)** | **RGB-Infrared** | ✓ | 51 252 | 338 839 | 5 |

Table 1: Statistical comparison among existing aerial detection datasets, 'F-OVOD' represent whether suitable for for open-vocabulary object detection, '-' indicates that the exact number was not specified in the original paper.

### 3.1 DATA ACQUISITION AND PROCESSING

The RGB-Infrared OVOD Dataset is built upon the DroneVehicle dataset. We retain only well-aligned RGB-infrared image pairs, removing image borders and edge-aligned object annotations from the original data. Each processed pair is resized to a standardized resolution of $512 \times 640$ pixels. In the original dataset, infrared and RGB images were annotated separately. To ensure consistency, we discard low-quality pairs with poor brightness and adopt horizontal bounding boxes from the infrared annotations only, converting them into COCO format. Following the approach in (Cheng et al., 2024), detection labels are transformed into textual category descriptions to support open-vocabulary detection. The dataset is split into *seen* and *unseen* categories: *seen* includes car, bus, truck, and van; *unseen* consists of freight car.

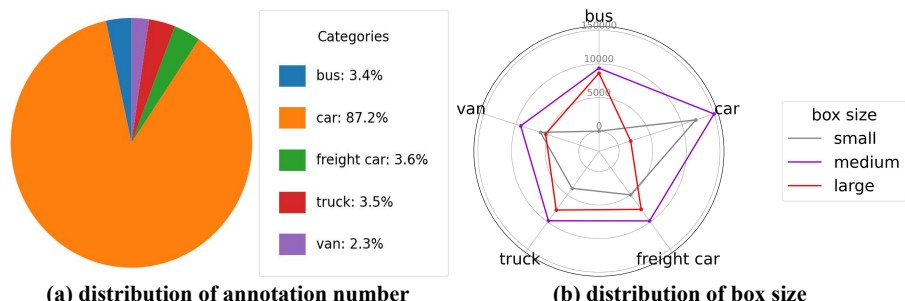

(a) distribution of annotation number      (b) distribution of box size

Figure 1: (a) Long-tail distribution of category annotations. (b) Box size distribution: small ($[1^2, 32^2)$), medium ($[32^2, 96^2)$), large ($[96^2, \infty)$). Most targets are small or medium-sized.

## 3.2 DATASET STATISTICS

The RGB-Infrared OVOD Dataset consists of 25,626 dual-modality image pairs covering five common object categories, with a total of 338,839 bounding boxes. As illustrated in Figure 1(a), the category "car" is the most frequent, accounting for 87%, while the remaining four categories are more evenly distributed and together make up 13%, indicating a long-tail distribution.

## 3.3 KEY FEATURES AND CHALLENGES OF THE DATASET

Our dataset has the following key features and associated challenges:

- **Diverse Scenarios**: The dataset includes both daytime and nighttime scenes, multiple viewing angles (15°, 30°, and 45°), and various flight altitudes (80 m, 100 m, and 120 m), capturing diverse real-world aerial conditions. Diverse scenarios resulting in changes in the imaging scale and projection relationship of the target, increasing the difficulty of feature extraction and recognition. For the same target at different height and viewing angles, there are large differences in appearance and contour, and the model needs to have generalization ability to deal with these changes.

- **Focus on Small and Medium Objects**: As illustrated in Figure 1(b),Over 90% of annotated objects are small or medium in size, highlighting the challenges of aerial small object detection (Lin et al., 2014). Moreover, the category annotations present a long - tail distribution. Such data imbalance is likely to affect the detection performance of models for targets of low - frequency categories.

- **Suitable for OVOD**: It is the first dataset for open-vocabulary object detection using aligned RGB-infrared image pairs.

## 4 METHOD

### 4.1 TASK DEFINITION

Our task is to perform open-vocabulary object detection using dual-modality images (RGB and infrared) from low-altitude scenes. The dataset is represented as $D = \{I_{ir}, I_{vi}, (b, y)r\}$, where $I_{ir} \in \mathbb{R}^{C \times H \times W}$ is the infrared image, $I_{vi} \in \mathbb{R}^{C \times H \times W}$ is the RGB image, and $b = \{x_{\min}, y_{\min}, w, h\} \in \mathbb{R}^4$ is the localization annotation, with $y$ being the category label. The dataset contains $r$ pairs of infrared and RGB images. Specifically, the dataset is divided into two parts: Seen classes $D_{seen}$ and unseen classes $D_{unseen}$, with the structure as described above. The seen and unseen classes have no overlap, and the corresponding word-level text annotations are $V_{seen}$ and $V_{unseen}$ respectively. The model is trained on images from both modalities in $D_{seen}$ along with their corresponding text annotations. It then identifies and localizes objects correctly in images from $D_{unseen}$ based on text prompts.

### 4.2 ARCHITECTURE OVERVIEW

Since the introduction of the YOLO-World model (Cheng et al., 2024), many similar lightweight open-vocabulary object detection methods have emerged, such as YOLO-Uniow (Liu et al., 2024a), Mamba-YOLO-World (Wang et al., 2025), DOSOD (He et al., 2024) and so on. Compared to large models like GroundingDINO, lightweight models have attracted significant interest for their efficient

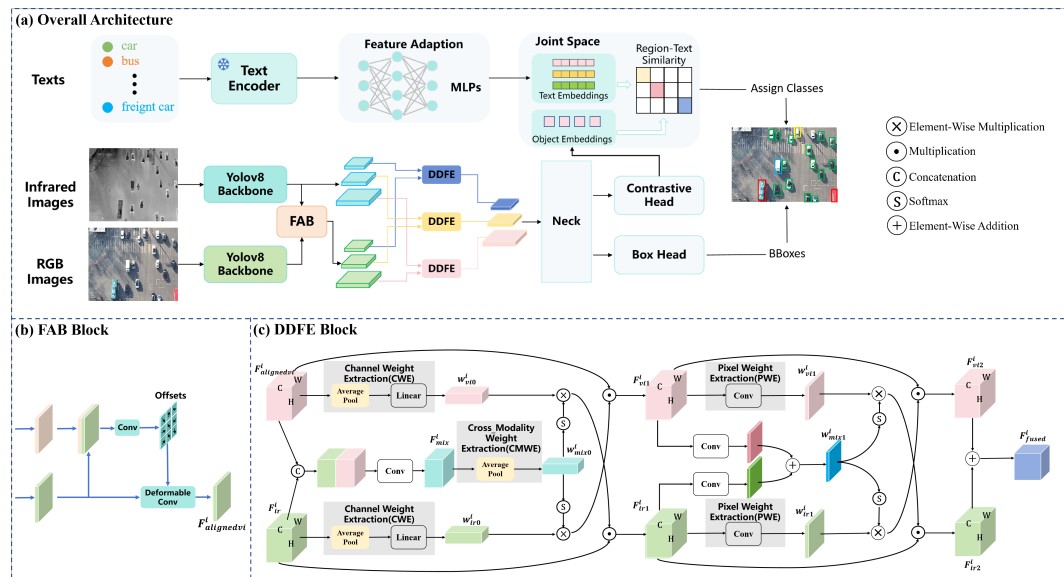

Figure 2: (a) Overall architecture of LSRI. It uses CLIP's text encoder to extract text features, which are then projected into the feature space via a text adapter. A two-branch image encoder is employed to extract features from RGB and infrared images. RGB features are aligned with infrared features through the FAB module, and then undergo the DDFE module for cross-modal enhancement and fusion. The target category is determined by calculating regional text similarity, and the bounding box is predicted by the bbox head. (b) Detailed structure of the Feature Alignment Block (FAB). (c) Detailed structure of the Dual-Dimension Feature Enhancement Block (DDFE).

performance and faster inference speed. Therefore, this paper builds upon the DOSOD architecture to further explore OVOD in low-altitude scenarios. As shown in Figure 2, LSRI follows the basic structure of DOSOD. For text feature extraction, we use a frozen CLIP text encoder and a text adaptor to project the text features into the joint space. For image feature extraction, we employ a dual-branch image encoder based on YOLOv8 to extract multi-level features from both infrared and RGB images. Since the original DroneVehicle dataset annotates infrared and RGB images separately, we use infrared image annotations as the unified labels for dual-modality images, but this results in a slight misalignment for the RGB images. To address this, we design a Feature Alignment Block (FAB) to align the RGB features with the infrared features, correcting this discrepancy. Next, we enhance both modality features along the channel and spatial dimensions using a Dual-Dimension Feature Enhancement Block(DDFE), followed by element-wise addition for feature fusion. Subsequently, as in a typical single-stage object detector, we employ a neck for feature refinement, and the bounding box head predicts class-agnostic bounding boxes, while the contrastive head extracts region features. In the joint space, the text features are aligned with the region features, and their similarity is computed to determine the target class. For details on the DOSOD model, please refer to the relevant literature (He et al., 2024) . Next, we will introduce the FAB and DDFE blocks we propose.

### 4.3 FEATURE ALIGNMENT BLOCK

The infrared image encoder is denoted as $E_{ir}$,and the RGB image encoder is denoted as $E_{vi}$. The features extracted from the last three blocks of the YoloV8 backbone are denoted as $F_{ir} = E_{ir}(I_{ir}) = \{F_{ir}^3, F_{ir}^4, F_{ir}^5\}$, $F_{vi} = E_{vi}(I_{vi}) = \{F_{vi}^3, F_{vi}^4, F_{vi}^5\}$. Features from different levels need to be aligned separately (Hu et al., 2025). As illustrated in Figure 2(b), the infrared and RGB features are concatenated along the channel dimension and passed through a convolutional layer to predict the offsets of the RGB features relative to the infrared features in both horizontal and vertical directions. Subsequently, deformable convolution is employed, utilizing the predicted offsets to spatially align the RGB features. The alignment process is formulated as follows:

$$\text{offset} = \text{Conv}(\text{Concat}(F_{ir}^i, F_{vi}^i)), \tag{1}$$

$$F_{alignedvi}^i = \text{DConv}(F_{vi}^i, \text{offset}). \tag{2}$$

where Concat() denotes concatenation along the channel dimension and $i \in \{3, 4, 5\}$ represents different feature level. The final aligned RGB features are $F_{alignedvi} = \{F_{alignedvi}^3, F_{alignedvi}^4, F_{alignedvi}^5\}$.

## 4.4 Dual-Dimension Feature Enhancement Block

To maximize the advantages of both infrared and RGB modalities, inspired by (Chen et al., 2024), we design the Dual-Dimension Feature Enhancement (DDFE) block, which enhances features along both channel and spatial dimensions. This helps preserve thermal radiation information from infrared images and detailed textures from RGB images, addressing the limitations of each modality, such as the sensitivity of RGB to illumination and the noise and low resolution in infrared images (Zhao et al., 2023; Huang et al., 2022; Zhao et al., 2025; Huang et al., 2025).

As shown in Figure 2(c), the DDFE consists of two main components. First, in the channel dimension, each single-modal feature is used to enhance the other modality in order to suppress redundant information. Specifically, the two features are concatenated along the channel dimension, and a convolution operation is applied to project the combined features back to their original size, thereby filtering out irrelevant channels and generating a hybrid feature $F_{mix}^i$. The Channel Weight Extraction (CWE) module extracts weights from individual modalities, while the Cross-Modality Weight Extraction (CMWE) module captures interactions between modalities. These modules are used to compute the corresponding channel-wise weights $w_{v0}^i, w_{ir0}^i, w_{mix0}^i \in \mathbb{R}^{b \times c \times 1 \times 1}$. After obtaining the importance scores of modal-specific and modal-mixed features across different channels, the mixed-modal weights are used to reassign the single-modal weights via element-wise multiplication. This process can be formulated as follows:

$$F_{mix}^i = \text{Conv}(\text{Concat}(F_{ir}^i, F_{alignedvi}^i)), \tag{3}$$

$$w_{vi0}^i = \text{CWE}(F_{alignedvi}^i), w_{ir0}^i = \text{CWE}(F_{ir}^i), \tag{4}$$

$$w_{mix0}^i = \text{CMWE}(F_{mix}^i), \tag{5}$$

$$w_{envi0}^i = w_{vi0}^i \odot \text{softmax}(w_{mix0}^i), \tag{6}$$

$$w_{enir0}^i = w_{ir0}^i \odot \text{softmax}(w_{mix0}^i), \tag{7}$$

$$F_{ir1}^i = F_{ir}^i \odot w_{envi0}^i, F_{vi1}^i = F_{alignedvi}^i \odot w_{enir0}^i. \tag{8}$$

where $F_{ir1}^i$ and $F_{ir1}^i$ denote the enhanced feature, $\odot$ denotes element-wise multiplication.

Next, within the spatial dimension, two single-modal features are employed to enhance the other modality. To fully capture the spatial characteristics of each modality, we design a module named PWE that extracts the pixel-wise importance weights of each single-modal feature($w_{v1}^i$ and $w_{ir1}^i$) along the spatial axis, we also extract cross-modal spatial-dimension pixel weights $w_{mix1}^i$. Finally, we get three weights $w_{v1}^i, w_{ir1}^i, w_{mix1}^i \in \mathbb{R}^{b \times 1 \times w \times h}$, thereby obtaining the significance of modal-specific and modal-mixed information at different pixel locations. By performing element-wise multiplication, the mixed-modal weights are leveraged to reassign the single-modal weights. This process can be formulated as follows:

$$w_{mix1}^i = \text{Conv}(F_{ir1}^i) + \text{Conv}(F_{vi1}^i), \tag{9}$$

$$w_{ir1}^i = \text{PWE}(F_{ir1}^i), w_{vi1}^i = \text{PWE}(F_{vi1}^i), \tag{10}$$

$$w_{envi1}^i = w_{vi1}^i \odot \text{softmax}(w_{mix1}^i), \tag{11}$$

$$w_{enir1}^i = w_{ir1}^i \odot \text{softmax}(w_{mix1}^i), \tag{12}$$

$$F_{ir2}^i = F_{ir1}^i \odot w_{envi1}^i, F_{vi2}^i = F_{vi1}^i \odot w_{enir1}^i, \tag{13}$$

Finally, the enhanced representations of the two modalities are fused to produce the ultimate extracted features.This process can be formulated as follows:

$$F_{fused}^i = F_{ir2}^i + F_{vi2}^i. \tag{14}$$

### 4.5 Loss Function

Inspired by DOSOD, we construct the region-text contrastive loss using region–text similarity and the cross-entropy between object–text assignments. Bounding-box regression is supervised by an IoU loss (Chen et al., 2020) together with a distributed focal loss. The overall training objective is defined as:

$$\mathcal{L} = \lambda_1 \mathcal{L}_{cls} + \lambda_2 \mathcal{L}_{iou} + \lambda_3 \mathcal{L}_{dfl}, \tag{15}$$

where $\lambda_1, \lambda_2, \lambda_3$ are loss weights.

Since the majority of objects in the dataset are small or medium-sized, we replace $\mathcal{L}_{iou}$ with $\mathcal{L}_{SAFit}$ Ying et al. (2025) , defined as:

$$L_{SAFit} = 1 - \text{SAFit}, \tag{16}$$

$$\text{SAFit} = \frac{1}{1 + e^{-(\sqrt{A}/C - 1)}} \times \text{IOU} + (1 - \frac{1}{1 + e^{-(\sqrt{A}/C - 1)}}) \times \text{NWD(C)}. \tag{17}$$

$$\text{NWD(K)} = \exp\left(-\frac{\sqrt{\text{W}_2^2(N_p^{\text{T}}, N_{gt})}}{K}\right). \tag{18}$$

where $\text{W}_2^2(N_p^{\text{T}}, N_{gt})$ is the Wasserstein distance between the predicted bounding box $N_p = [cx_p, cy_p, w_p/2, h_p/2]$ and the ground truth bounding box $N_{gt} = [cx_{gt}, cy_{gt}, w_{gt}/2, h_{gt}/2]$. The coordinates of the center points of $N_p$ and $N_{gt}$ are $(cx_p, cy_p)$, the width is $w$, and the height is $h$. $k$ is a hyperparameter closely related to the dataset. $A$ is the area of the GT bounding box, and $C$ is a constant used to balance the metrics of NWD (Wang et al., 2021) and IoU in a size - aware manner.

## 5 Experiment

### 5.1 Datasets and Evaluation Benchmarks

The RI-OVOD dataset was used in the experiments, with all data annotated using horizontal bounding boxes. Models were trained on seen classes data and validated on unseen classes. Evaluation metrics included $mAP$, $AP_{50}$, $AP_{75}$.

### 5.2 Implementation Details

For comparative experiments, we selected small and medium versions of four methods: YOLO-Uniow, YOLO-World, Mamba-YOLO-World and DOSOD. Three types of experiments were conducted: open-vocabulary object detection (OVOD) on RGB images, OVOD on infrared images, and OVOD on dual-modal (RGB-infrared) images. In experiments utilizing dual-modal images for training, to ensure fairness, other methods were adjusted to a model architecture capable of supporting dual-modal training in accordance with LSRI's approach, thereby verifying the superiority of the proposed method.

All experiments were based on the YOLO-World codebase, with dependencies on MMYOLO and MMDetection. Since the original code did not support simultaneous loading of infrared and RGB images, we reimplemented a data loader capable of reading text and dual-modal images, along with corresponding data augmentation functions. Experiments were performed on 4 NVIDIA RTX 4090 D GPUs. For our proposed method, the CLIP text encoder was frozen during training to extract text features. The batch size was set to 12, using the AdamW optimizer with a learning rate of 2e-3, and training ran for 50 epochs. Due to memory constraints, no data augmentation was applied when training with dual-modal data. The hyperparameters $\lambda_1, \lambda_2, \lambda_3$ are set to 0.5, 7.5, and 0.375 respectively, and the parameter $C$ is set to 32, the optimal validation result from the 50 epochs was selected as the final evaluation metric.

### 5.3 Comparative Experiment

The main goals of this experiment are to validate the following two hypotheses: first, training with multimodal data can effectively improve the performance of open-vocabulary object detection models compared to single-modal data; and second, our proposed LSRI method outperforms other existing methods.

| Method | data type | RGB-Infrared OVOD Dataset | | |
|---|---|---|---|---|
| | | mAP | AP$_{50}$ | AP$_{75}$ |
| *Small Model* | | | | |
| YOLO-Worldv1_s Cheng et al. (2024) | RGB | 0.084 | 0.179 | 0.063 |
| | Infrared | 0.115 | 0.211 | 0.116 |
| | RGB-Infrared | 0.108 | 0.221 | 0.092 |
| YOLO-Uniow_s Liu et al. (2024a) | RGB | 0.110 | 0.224 | 0.096 |
| | Infrared | 0.145 | 0.253 | 0.146 |
| | RGB-Infrared | 0.129 | 0.236 | 0.129 |
| Mmaba-YOLO-World_s Wang et al. (2025) | RGB | 0.110 | 0.230 | 0.092 |
| | Infrared | 0.129 | 0.246 | 0.124 |
| | RGB-Infrared | 0.133 | 0.242 | 0.135 |
| DOSOD_s He et al. (2024) | RGB | 0.130 | 0.256 | 0.118 |
| | Infrared | 0.160 | 0.281 | 0.169 |
| **LSRI_s(ours)** | RGB-Infrared | **0.170** | **0.303** | **0.170** |
| *Medium Model* | | | | |
| YOLO-Worldv1_m Cheng et al. (2024) | RGB | 0.089 | 0.184 | 0.075 |
| | Infrared | 0.132 | 0.228 | 0.140 |
| | RGB-Infrared | 0.118 | 0.226 | 0.114 |
| YOLO-Uniow_m Liu et al. (2024a) | RGB | 0.102 | 0.210 | 0.086 |
| | Infrared | 0.169 | 0.288 | 0.182 |
| | RGB-Infrared | 0.109 | 0.209 | 0.105 |
| Mmaba-YOLO-World_m Wang et al. (2025) | RGB | 0.098 | 0.205 | 0.083 |
| | Infrared | 0.176 | 0.302 | 0.188 |
| | RGB-Infrared | 0.142 | 0.277 | 0.135 |
| DOSOD_m He et al. (2024) | RGB | 0.121 | 0.246 | 0.105 |
| | Infrared | 0.164 | 0.302 | 0.163 |
| **LSRI_m(ours)** | RGB-Infrared | **0.205** | **0.356** | **0.214** |

Table 2: Performance comparison of different methods on the RGB-Infrared OVOD Dataset. Small and medium models are analyzed separately, and the performance of each method across three experimental scenarios is compared.

### 5.3.1 MULTIMODAL DATA ANALYSIS

In the single-modal experiments, the results show that object detection using infrared images alone outperforms the RGB image model. This phenomenon can be explained by the fact that infrared images can compensate for the limitations of RGB images in poor lighting and weather conditions. Especially in low-light environments, infrared images provide more structural information, enhancing the detection performance. In the open-vocabulary object detection experiments with multimodal images, compared with training on single-modal data, our proposed LSRI demonstrates that combining RGB and infrared images as training data can significantly improve model performance. However, for methods like YOLO-World and Mamba-YOLO-World, although there was some improvement in multimodal data training, the performance did not significantly increase. In some cases, the performance was even slightly worse than the best results from single-modal image experiments. This may be because these methods had already been optimized for single-modal data during training, and the introduction of multimodal data did not effectively enhance the model's learning capability. It may also have increased the model's complexity, leading to a negative impact.

### 5.3.2 METHOD SUPERIORITY ANALYSIS

As shown in Table 2, both the small and medium models of our proposed LSRI achieve state-of-the-art performance in all experiments. On the RGB-Infrared OVOD dataset, the LSRI method outperforms all other methods in all evaluation metrics. From the visualization results in Figure 3, our method remains the best. This superiority can be attributed to the innovative module design and training strategy proposed in LSRI. By combining the strengths of both RGB and infrared images, we have enhanced the model's robustness to various visual features through joint learning. In contrast, although other methods have attempted to use multimodal data to some extent, they did not fully exploit the complementary nature of the two modalities, leading to less significant improvements in their performance.

### 5.4 ABLATION EXPERIMENT

To validate the effectiveness of the proposed method, we conducted a series of ablation experiments using LSRI_s on the dataset we constructed. These experiments focused on three main aspects: (1) the impact of the loss functions; (2) the contribution of the FAB; and (3) the influence of the DDFE module for infrared-RGB feature fusion. Before using the DDFE module, we only used simple

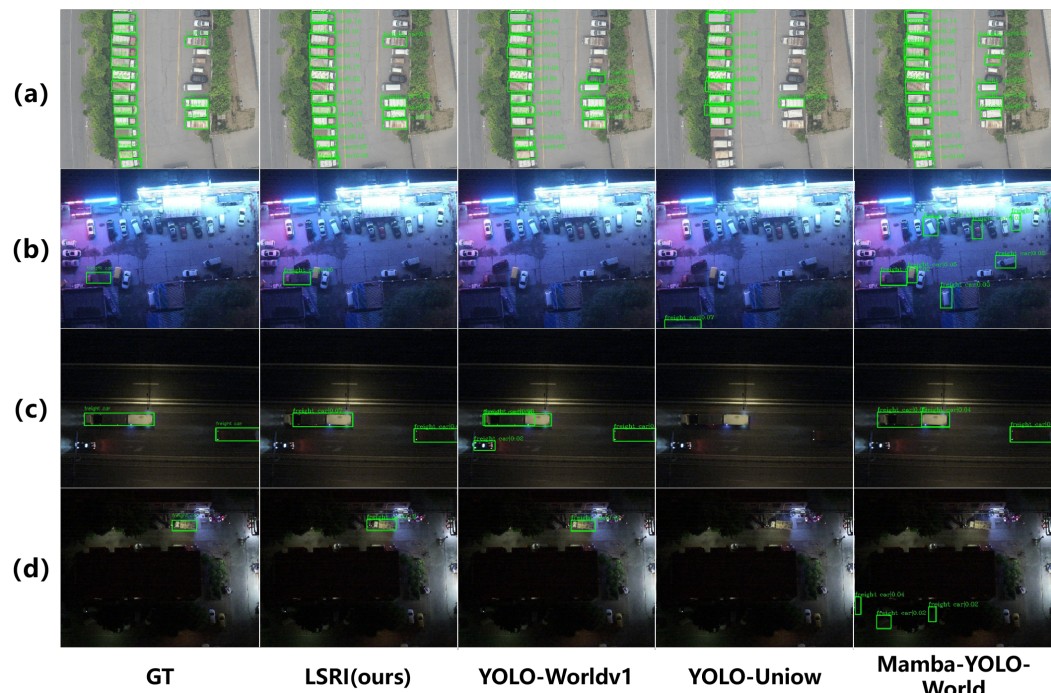

|  | GT | LSRI(ours) | YOLO-Worldv1 | YOLO-Uniow | Mamba-YOLO-World |

Figure 3: Visualization results. Figures (a) to (d) show the detection results under different lighting conditions and scenarios using the prompt "freight car" and the corresponding RGB and infrared images. Compared with our method, other methods exhibit false detections and missed detections.

| IoU | SAFit | FAB | ADD | DDFE | mAP | AP$_{50}$ | AP$_{75}$ |
|-----|-------|-----|-----|------|-----|-----------|-----------|
| √ | | | √ | | 0.189 | 0.333 | 0.197 |
| | √ | | √ | | 0.197↑ | 0.342↑ | 0.201↑ |
| | √ | √ | √ | | **0.208↑** | **0.355↑** | **0.219↑** |
| | √ | √ | | √ | 0.170↓ | 0.303↓ | 0.170↓ |

Table 3: Performance metrics with different loss and training strategies.

feature addition to achieve the fusion of infrared and RGB features. The quantitative results are shown in Table 3.

From the experimental results, $L_{SAFit}$ significantly contributes to performance improvements across multiple metrics. Furthermore, the addition of the FAB also leads to consistent gains. The introduction of the DDFE module did not lead to performance improvements. Compared to simple feature addition, the complex feature enhancement and fusion strategy resulted in a performance drop. However, it still outperformed models trained with either single modality alone. This suggests that deeply nested and non-linear fusion structures can easily introduce excessive noise or unstable gradients under limited training data, making optimization more difficult.

## 6 CONCLUSION

This paper constructs the RGB-Infrared OVOD dataset, the first open-vocabulary object detection dataset designed for low-altitude scenarios using aligned multimodal data, providing a solid data foundation for open-set aerial detection. Meanwhile, we propose the LSRI architecture, the first model that leverages aligned infrared and RGB images for open-vocabulary object detection, and its effectiveness has been demonstrated through extensive experiments. In future work, we will further explore the fusion of infrared and RGB features with textual features to improve the model's performance. We believe that the development of larger-scale datasets and more advanced foundational models will greatly accelerate the progress of open-set aerial detection.

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

## A  APPENDIX

You may include other additional sections here.

