# OpenReview forum: "Open-Vocabulary Object Detection for Low-Altitude Scenarios Using RGB-Infrared Data: A Benchmark and A New Method"
_ICLR.cc/2026/Conference — ICLR 2026 Conference Withdrawn Submission_

### Official Review · Reviewer_2NUK · 2025-10-26

**Soundness:** 2
**Presentation:** 3
**Contribution:** 2
**Rating:** 2
**Confidence:** 4

**Summary:**

This paper presents RI-OVOD, the first open-vocabulary RGB–infrared aligned object detection dataset for low-altitude scenarios, containing 25,626 paired images and annotations for five object categories. Based on this dataset, the authors propose a multimodal detection model named LSRI, which integrates RGB texture and infrared thermal information through a Feature Alignment Block (FAB) and a Dual-Dimensional Feature Enhancement (DDFE) module. In addition, a size-adaptive loss function (LSAFit) is introduced to improve small-object detection performance. Experiments show that LSRI achieves an AP₅₀ of 0.356 on RI-OVOD, significantly outperforming single-modality baselines.

**Strengths:**

# **Strengths**

1. **Originality:** This paper introduces **RI-OVOD**, the first open-vocabulary RGB–infrared aligned object detection dataset designed for low-altitude scenarios, filling an existing research gap in multimodal data for such environments. The proposed **LSRI** model incorporates the **Feature Alignment Block (FAB)** and **Dual-Dimensional Feature Enhancement (DDFE)** modules to enhance multimodal feature representation, demonstrating a meaningful level of innovation.

2. **Quality:** The experimental design is rigorous, featuring comprehensive comparisons based on the RI-OVOD dataset that cover both single-modality and multimodal settings. The LSRI model achieves a statistically significant performance improvement (AP₅₀ from 0.302 to 0.356) on small- and medium-scale models. The data preprocessing and model optimization processes are described in sufficient technical detail to support reproducibility.

3. **Clarity:** The paper is clearly structured, with the introduction effectively outlining the motivation and background. The methodology section is detailed, and Figure 2 provides an intuitive overview of the overall architecture and key components, aiding readers in understanding the model’s operation.

4. **Significance:** This work has strong potential for practical applications in UAV remote sensing and low-altitude perception scenarios. It contributes to advancing open-vocabulary detection toward multimodal remote sensing and small-object detection, offering both academic and real-world significance.

**Weaknesses:**

# **Weaknesses**

1. **Limited Openness in Dataset Design:**
   The “open-vocabulary” setting of the RI-OVOD dataset is overly simplistic. It includes only one unseen category, *freight car*, while all seen categories (*car, bus, truck, van*) and the unseen one belong to the same high-level semantic group — vehicles. This narrow semantic scope weakens the evaluation of the model’s true generalization ability. A genuinely open-vocabulary model should be tested on objects with larger semantic divergence from the training categories. The reported good performance may thus reflect fine-grained vehicle classification rather than true open-vocabulary detection.

2. **Questionable Effectiveness of the DDFE Module:**
   The ablation results (Table 3) show that introducing the DDFE module actually *degrades* detection performance. As one of the main claimed contributions, the DDFE’s negative impact directly contradicts the paper’s stated benefits. Without a clear explanation or redesign, this undermines the claimed novelty and effectiveness of the proposed architecture.

3. **Fairness of Baseline Comparisons:**
   The paper mentions modifying baseline models to enable bimodal (RGB–infrared) input, but does not specify *how* these modifications were implemented. Since multimodal fusion strategy significantly affects performance, a simple early fusion or concatenation for baselines — compared to a carefully designed FAB+DDFE fusion in LSRI — may lead to unfair advantages. The paper should clarify the fusion approach used in baselines to ensure a fair comparison.

4. **Insufficient Analysis of Text–Vision Alignment:**
   The CLIP text encoder is frozen, and the design and role of the text adaptor layer are underexplored. The paper lacks discussion of how different text adaptation strategies, languages, or prompt formulations influence model performance. Adding sensitivity experiments on textual prompts would strengthen the analysis.

5. **Lack of Computational Efficiency Evaluation:**
   The paper does not report inference speed (FPS), parameter count, or memory consumption. Without such measurements, it is difficult to assess the computational cost and practical deployability of the LSRI model.

6. **Insufficient Validation of Generalization:**
   The evaluation only includes detection results for the unseen category (*freight car*) without showing results for seen or all categories, which diverges from standard practice in the OVOD field. Furthermore, the experiments are limited to the self-constructed RI-OVOD dataset. To demonstrate generalization, the authors should test the model on additional open-vocabulary or multimodal detection datasets.

**Questions:**

# **Questions**

1. **Regarding the DDFE Module:**
   Given that the ablation experiments indicate a *performance drop* after introducing the DDFE module, why do the authors still highlight it as one of the paper’s main contributions? Beyond the explanation of data limitation and model complexity, was there any deeper analysis (e.g., examining feature redundancy or training instability)? Have the authors tested simplified versions or alternative fusion strategies? If the best-performing model does not include this module, should the contribution claims be revised accordingly?

2. **On the Choice of Unseen Categories:**
   The current “open-vocabulary” setting uses *freight car* as the only unseen category, which is semantically close to the seen classes. Have the authors considered experimenting with more challenging seen/unseen splits, such as treating *bus* or *truck* as unseen categories, or even including objects from completely different domains (e.g., synthetic or aerial non-vehicle classes) to better validate the generalization ability of LSRI?

3. **Adaptation of Baseline Models:**
   The paper mentions modifying baselines like YOLO-World to support bimodal RGB–infrared input. Could the authors elaborate on how exactly these modifications were made? Specifically, at which network stage are the two modalities fused, and what type of fusion strategy (e.g., early, mid-level, or late fusion) is used? Such details are critical for evaluating the fairness of baseline comparisons.

4. **Impact of Data Augmentation:**
   Have the authors analyzed how removing data augmentation during single-modality (especially RGB) training affects baseline performance? This would clarify whether the “no augmentation” setting biases the comparison in favor of multimodal models. Additionally, if computational resources permit, how would LSRI perform when trained with standard data augmentation?

5. **Prompt Design and Text Robustness:**
   Please discuss the sensitivity of detection performance to different text prompts. For instance, how does the model respond to synonym prompts such as *“freight car”* vs. *“cargo truck”*? Have the authors considered using more descriptive or sentence-level prompts (e.g., “a truck carrying goods”) to test robustness in real-world text conditions?

6. **Computational Efficiency and Real-Time Performance:**
   As bimodal processing typically increases inference latency, what is the actual runtime efficiency of LSRI? Please provide a comparison of FPS, parameter count, and FLOPs with baseline models to demonstrate the model’s practicality and computational trade-offs.

---

### Official Review · Reviewer_rr8E · 2025-10-29

**Soundness:** 2
**Presentation:** 2
**Contribution:** 2
**Rating:** 2
**Confidence:** 4

**Summary:**

This work addresses open-vocabulary object detection (OVOD) in low-altitude UAV scenarios by proposing a multimodal approach using aligned RGB-Infrared imagery. The paper introduces RI-OVOD, the first open vocabulary aerial object detection dataset with aligned RGB-Infrared image pairs, containing 25,626 pairs with vocabulary-level annotations. Building on this dataset, the authors propose LSRI, a novel architecture that leverages complementary information from both modalities through a Feature Alignment Block (FAB) and a Dual-Dimension Feature Enhancement (DDFE) module to improve detection performance, particularly for small objects prevalent in low-altitude scenarios. However, the experimental results fail to effectively support the authors' claims.

**Strengths:**

The motivation is reasonable

**Weaknesses:**

1).The authors claim RI-OVOD is "the first" open-vocabulary dataset using aligned RGB-Infrared images, but existing datasets like DroneVehicle and RGB-Tiny can also be converted to open-vocabulary format by treating category names as text annotations. What is the difference between RI-OVOD and these datasets?

2)With only 5 categories (car, bus, truck, van, freight car), and the dataset fails to cover common aerial object categories comprehensively. How can such limited category coverage demonstrate open vocabulary detection capability?

3)The paper splits the dataset into base (seen) and novel (unseen) classes, which is standard for open vocabulary evaluation. However, the experimental results in Table 2 do not report separate performance on base vs. novel classes.

4)No visualization or ablation study demonstrates that FAB actually corrects the claimed misalignment.

5)The ablation study (Table 3) clearly shows that DDFE decreases performance compared to simple feature addition. Should a method that underperforms a trivial baseline (element-wise addition) be published as a technical contribution?

6)The ablation study reveals that the loss function (LSAFit) is the primary contributor to performance improvement. However, LSAFit appears to be a direct application of existing work. What is the novel contribution of the loss function?

7)Mismatched Citation and Context: For example, Lines 43-46, the paper states: "However, most existing OVOD methods are trained and tested on natural images [1], and due to the significant differences between datasets in different domains, these methods struggle to generalize effectively to other tasks in aerial object detection [2]." However, both cited references [1] and [2] are actually works focusing on aerial imagery, not natural images:

[1] Zang Z, Lin C, Tang C, et al. Zero-shot aerial object detection with visual description regularization[C]//Proceedings of the AAAI Conference on Artificial Intelligence. 2024, 38(7): 6926 6934.
 [2] Pan J, Liu Y, Fu Y, et al. Locate anything on earth: Advancing open-vocabulary object detection for remote sensing community[C]//Proceedings of the AAAI Conference on Artificial Intelligence. 2025, 39(6): 6281-6289

**Questions:**

Please refer to the weaknesses section

---

### Official Review · Reviewer_mv1n · 2025-10-31

**Soundness:** 2
**Presentation:** 2
**Contribution:** 2
**Rating:** 4
**Confidence:** 4

**Summary:**

This paper presents RI-OVOD, a new benchmark and baseline for object detection in low-altitude UAV (drone) imagery using aligned RGB–Infrared (IR) inputs.
The authors claim this as the first open-vocabulary object detection (OVOD) dataset and model for multimodal low-altitude scenes. The work consists of two main parts:
-  RI-OVOD (RGB-Infrared Open-Vocabulary Object Detection): Derived from the DroneVehicle dataset (Sun et al., 2022), containing 25,626 aligned RGB–IR pairs and ~340K bounding boxes. Covers five vehicle-related categories: car, bus, truck, van, and freight car, with the last held out as an “unseen” class to simulate open-vocabulary detection. Includes textual category descriptions and long-tail statistics
- LSRI (Low-Altitude Scene RGB-Infrared model).

On RI-OVOD, LSRI achieves 0.356 AP50, outperforming single-modality baselines (0.302 for IR-only, 0.246 for RGB-only). The method also surpasses YOLO-World, YOLO-UniOW, Mamba-YOLO-World, and DOSOD on all evaluated metrics.

**Strengths:**

- New benchmark: Provides the first aligned RGB–IR dataset for low-altitude UAV object detection with text annotations.
- Multimodal fusion: The FAB alignment and LSAFit loss address important issues (modality mismatch and small-target bias).
- Empirical gains: Consistent performance improvement over single-modal and prior OVOD models.

**Weaknesses:**

- Not truly open-vocabulary: The dataset includes only five classes, with freight car treated as the sole “unseen” category.
- Limited semantic scope: All categories are vehicles; there is no variation in object semantics, scene types, or linguistic expressions.
- Ablation inconsistencies: Table 3 shows that DDFE slightly decreases performance. Can author discuss more?

**Questions:**

- Why was freight car chosen as the only unseen class? Have you tested any cross-dataset transfer?
- Could you provide per-category AP scores to better show zero-shot generalization?
- Would you consider repositioning the contribution as “multimodal UAV detection” rather than “open-vocabulary”?

---

### Official Review · Reviewer_M1GP · 2025-11-02

**Soundness:** 2
**Presentation:** 3
**Contribution:** 2
**Rating:** 4
**Confidence:** 5

**Summary:**

This paper introduces RIOVOD, the first RGB-Infrared dataset for low-altitude Open-Vocabulary Object Detection (OVOD). It also proposes the LSRI model, a dual-branch architecture based on DOSOD, which claims to use a DDFE module for feature fusion. Experiments show that fusing multimodal data is superior to using single-modality data.

**Strengths:**

The main strength of the paper is the contribution of the RIOVOD dataset, the first benchmark for low-altitude multimodal OVOD. Furthermore, the paper validates the effectiveness of using the LSAFit loss and the FAB alignment module.

**Weaknesses:**

1. The primary weakness lies in the failure of its core methodological innovation, the DDFE module. The ablation study (Table 3) clearly demonstrates that DDFE performs worse than a simple baseline (ADD), which invalidates the paper's claim of methodological novelty.
2. The paper's narrative is misleading. It continually emphasizes the DDFE module, yet the state-of-the-art (SOTA) results (Table 2) likely stem from a simpler model that does not incorporate DDFE. This approach lacks rigor.
3. The model's reliance on late fusion (DDFE/ADD) may overlook crucial, low-level feature interactions between RGB and infrared data, especially when one modality is of low quality.
4. Unreasonable OVOD Setup and Insufficient Motivation for Using OVOD: The OVOD setup provided by the authors is unreasonable, and the motivation for introducing OVOD is insufficient. OVOD models typically require large-scale text-image semantic alignment training and multiple classes for testing, in order to repeatedly evaluate the model's open-vocabulary capabilities. However, RIOVOD only contains five categories, with only one class used for testing, which raises the question of why OVOD is necessary here. Additionally, the "unseen" class in the paper is also a vehicle, which is not much different from the known classes, thus failing to demonstrate the open-vocabulary capability.
5. The experiments are not complete. The authors need to provide performance comparisons for seen classes, unseen classes, and all classes combined. Additionally, they should include more qualitative and quantitative analyses of various fusion modules and loss functions. This should include comparisons with other common fusion modules and an examination of how varying the weights of the loss functions affects the results.
6. YOLO-World incorporates text features in the neck of its architecture, whereas the authors of this paper only utilize text features in the head for contrastive classification. It is unclear whether the structure proposed by the authors can be adapted to other open-vocabulary detectors. Additionally, other models generally perform worse when using RGB-I image types, yet the authors did not investigate the reasons behind this decline in performance. The authors should provide a more comprehensive approach to handling RGB-I image types and offer insights on how to enhance performance in this context.

**Questions:**

1. Why does the DDFE module cause significant performance degradation? If experiments show that it is ineffective, why is it still presented as the core innovation of the paper?
2. Is the state-of-the-art (SOTA) result in Table 2 derived from the model that includes DDFE, or is it from the simpler 'ADD' baseline?
3. The 'unseen' category ('freight car') is semantically very similar to the 'seen' categories ('car, bus, truck, van'). Does this relatively small semantic gap truly test the intended "open-vocabulary" generalization, or is it merely about fine-grained sub-category detection?
4. The LSAFit loss employs NWD to handle small objects. Could the authors provide an ablation study isolating the contribution of NWD compared to other robust IoU-based losses (e.g., DIoU or CIoU) to justify its selection in this specific scenario?
5. The Feature Alignment Block (FAB) is used to correct spatial misalignment. Could the authors provide visualizations (e.g., feature maps before and after FAB) or quantitative analyses to demonstrate the degree of misalignment in the dataset and how effectively the FAB resolves it?

---

### Note · Authors · 2025-12-03

I have read and agree with the venue's withdrawal policy on behalf of myself and my co-authors.